# Modified Keystone Perforator Island Flap Techniques for Small- to Moderate-Sized Scalp and Forehead Defect Coverage: A Retrospective Observational Study

**DOI:** 10.3390/jpm13020329

**Published:** 2023-02-15

**Authors:** Byung-Woo Yoo, Kap-Sung Oh, Junekyu Kim, Hyun-Woo Shin, Kyu-Nam Kim

**Affiliations:** Department of Plastic and Reconstructive Surgery, Kangbuk Samsung Hospital, Sungkyunkwan University School of Medicine, 29, Saemunan-ro, Jongno-gu, Seoul 03181, Republic of Korea

**Keywords:** forehead defect coverage, scalp defect coverage, keystone perforator island flap, reconstructive surgery

## Abstract

We aimed to demonstrate the effective application of keystone perforator island flap (KPIF) in scalp and forehead reconstruction by demonstrating the authors’ experience with modified KPIF reconstruction for small- to moderate-sized scalp and forehead defects. Twelve patients who underwent modified KPIF reconstruction of the scalp and forehead from September 2020 to July 2022 were enrolled in this study. In addition, we retrospectively reviewed and evaluated the patient’s medical records and clinical photographs. All defects (size range, 2 cm × 2 cm to 3 cm × 7 cm) were successfully covered using four modified KPIF techniques (hemi-KPIF, Sydney Melanoma Unit Modification KPIF, omega variation closure KPIF, and modified type II KPIF) with ancillary procedures (additional skin grafts and local flaps). All flaps (size range, 3.5 cm × 4 cm to 7 cm × 16 cm) fully survived, and only one patient developed marginal maceration that healed with conservative management. Furthermore, through the final scar evaluation with the patient satisfaction survey and Harris 4-stage scale, all patients were satisfied with their favorable outcomes at the average final follow-up period of 7.66 ± 2.14 months. The study showed that the KPIF technique with appropriate modifications is an excellent reconstructive modality for covering scalp and forehead defects.

## 1. Introduction

The scalp and forehead are unique dome-shaped components of the upper part of the head, which covers the cranium and the underlying brain [1]. Their underlying structures are similar despite regional differences affecting the reconstruction approach [1]. The tissue tautness caused by the convexity of the scalp and forehead frequently results in challenging defects that can be difficult to repair [1,2,3,4]. Primary closure is usually possible only for small scalp and forehead defects with a diameter of approximately 2–3 cm [1,5]. Diverse scalp and forehead reconstruction options have been developed, including skin grafting, dermal substitutes, tissue expander, locoregional flaps, and microvascular tissue transfer [1,2,3,4]. Skin grafting and dermal substitutes can be reliable for any superficial defect because of the rich blood supply to the scalp and forehead [6]. Flap surgery is essential for adequate coverage of deep defects where the underlying structures are exposed [6]. The local flap technique is typically helpful for covering small- to moderate-sized defects. Moreover, microvascular tissue transfer can be used for large-to-extensive defect coverage of the scalp and forehead [7,8]. Therefore, the free flap technique is the best reconstruction approach, particularly when covering areas with exposed alloplastic materials and neurocranial structures [8].

Many local flap techniques, such as advancement, rotation, transposition, rhomboid, and perforator flaps, have been devised to cover scalp and forehead defects with good functional and aesthetic outcomes [1,2,3,4,6,8]. Paradoxically, there is no unrivaled local flap for scalp and forehead reconstruction. Among the various types of local flap techniques, the keystone perforator island flap (KPIF), devised from Behan’s experience in over 300 cases with successful outcomes in 2003, has unique features, including intuitional defect-adaptable design and remarkable reproducibility with a minimal learning curve [6,7,9,10,11]. Based on these features, KPIF has become a popular reconstructive modality in most parts of the human body [9,10,11]. Despite this popularity, studies applying KPIF to scalp and forehead reconstruction are limited compared with other areas, such as the face, trunk, and extremities [9,10,11,12]. Moreover, a previous study reported that the scalp is a relative contraindication to KPIF reconstruction [13]. Therefore, we presented modified KPIF techniques to cover small- to moderate-sized scalp and forehead defects based on our experience. This study aimed to facilitate the effective application of KPIF with scalp and forehead defect coverage modifications to enhance the potential utility of KPIF reconstruction in plastic surgery.

## 2. Materials and Methods

The study protocol and research procedures complied with the ethical guidelines of the 1975 Declaration of Helsinki, and the Institutional Review Board of Kangbuk Samsung Hospital approved this study (approval number: 2022-10-012). Written consent was obtained from all patients for information and images from online open-access publications before performing treatment procedures and surgeries.

In this study, we enrolled patients with scalp and forehead defects that were covered using modified KPIF techniques from September 2020 to July 2022. Patients who underwent scalp and forehead defect coverage using other local flap techniques or skin grafts without the KPIF technique were excluded. We retrospectively reviewed and evaluated the patients’ data from their electronic medical records and clinical images, including the cause of defects, defect location, defect size, flap location, flap size, flap type, ancillary procedures, flap survival, postoperative complications, final scar appearance, and follow-up periods. Data from all patients were collected, processed, and analyzed using Microsoft Excel (Microsoft, Redmond, WA, USA) in an anonymized state.

### 2.1. Surgical Techniques

In cases of defects, which result from non-oncologic causes, such as trauma, postoperative wound necrosis, and wound infection, patients received preoperative wound preparation management that included empirical antibiotic treatment, serial wound debridement, and wound dressing for approximately 1–2 weeks [7,11]. After sufficient wound preparation, final debridement followed by KPIF reconstruction was performed. In cases of defects resulting from oncologic causes, such as skin malignancy, we performed wide excision with a safety margin, followed by KPIF reconstruction.

Our senior author performed the KPIF operation in either the supine position for forehead and anterior scalp defects or the prone position for posterior scalp defects under general or local anesthesia, in accordance with the defect size and each patient’s general condition. After complete debridement or wide excision, the final defect size was measured, and the flap was designed. When designing the flap, we considered two important factors as follows: defect width and laxity of the surrounding tissue. The flap was designed to be in an area with sufficient tissue laxity, and the flap width was designed to be 1.5–2 times larger than the defect width. However, we did not use a handheld Doppler device for detecting perforator hotspots because the scalp and forehead have abundant vascularity as perforator-rich areas. Four modifications of the KPIF were used in this study as follows: hemi-KPIF, which includes skin incision and division of the deep fascia at the unilateral apex with more than one-third of the ipsilateral-sided outer curvilinear line [14]; the Sydney melanoma unit modification (SMUM) KPIF, which involves the maintenance of a skin bridge along the outer curvilinear line [15]; the omega variation closure (OVC) KPIF, which includes fish-mouth-shaped defect closure through the addition of rotational flap movement [16]; and the modified Type II KPIF, comprising the deep fascia division along the whole circumference line of the flap [10,11]. Figure 1 illustrates these four KPIF modifications [11]. The choice of modification used was made intraoperatively [11,17]. In cases of the defect coverage with only the hemi-KPIF, skin incision and the deep fascia (galea aponeurotica and temporoparietal fascia) were divided at the one-sided apex with more than one-third of the outer curvilinear line. In cases where only the hemi-KPIF was judged insufficient for the defect, we created the SMUM KPIF, which involved another skin incision and deep fascia division at the other-sided apex while maintaining a skin bridge along the outer curvilinear line. If flap vascularity was stable and sufficient, skin incision and deep fascia division continued over the remaining skin bridge to form the modified Type II KPIF. After minimal undermining of the flap margin and meticulous bleeding control, flap inserting was performed as follows: first, the defect-sided flap was sutured by either linear closure or OVC (in cases where additional flap movement was required to lessen closure tension); second, V-Y advancement closure was made at either the unilateral (in the case of the hemi-KPIF) or bilateral apexes (in the case of other modifications) of the flap; and lastly, the repair of the donor site was performed [11]. If the donor site was not covered by primary closure or the remaining defect was not entirely covered by these modified KPIF techniques, ancillary procedures such as skin grafting and other local flaps were additionally used. After completing all the procedures, mild compressive dressings with foam materials were applied.

### 2.2. Evaluation of Final Scar Appearance

At the final follow-up, each patient was requested to rate their subjective satisfaction with the final scar appearance on a scale of 1 to 10. In addition, three independent plastic surgeons were asked to rate the objective final scar appearance following the Harris 4-stage scale as excellent, good, fair, or poor [18,19].

## 3. Results

Table 1 summarizes the clinical data and patient characteristics. Overall, 12 patients (9 men and 3 women) aged 58–84 years (average age ± standard deviation, 71.91 ± 7.44 years) were included. The etiology of the defects included skin necrosis after skin avulsion injury, wide local excision of basal cell carcinoma, skin necrosis after craniotomy, and cellulitis resulting from a ruptured epidermoid cyst in five, three, two, and two patients, respectively. The defect locations included the scalp in nine patients (frontal scalp in four, vertex scalp in two, and occipital scalp in three) and the forehead in three (central and lateral forehead in one, lateral forehead in one, and temporal forehead in one). The defect and flap sizes ranged from 2 cm × 2 cm to 3 cm × 7 cm and 3.5 cm × 4 cm to 7 cm × 16 cm, respectively. Hemi-KPIF was used in four cases (three scalp and one forehead defects), SMUM and OVC KPIF were used in four cases (three scalp and one forehead defects), modified Type II KPIF was used in three defects (two scalps and one forehead defects), and double-opposing hemi-KPIF was used in one scalp defect. Five patients required ancillary procedures to cover the donor site and the remaining defect. Skin graft for donor site closure and that for the remaining defect coverage was performed in two patients each, and rotational flap for remaining defect coverage was performed in one patient. No flap-related complications or full flap survival in any of the cases were observed. One case (case 7) showed marginal maceration of the 3-point suture area of the donor site, which was a partial-depth dehiscence with an incompletely healed wound margin. However, it was alleviated with conservative wound dressing and was completely healed without further surgical management. No other postoperative complications were observed in any of the other cases. Furthermore, the outcomes after an average follow-up period of 7.66 ± 2.14 months (range, 4–10 months) were subjectively (average patient satisfaction score, 8.16 ± 0.71) and objectively (the Harris 4-stage scale, more than fair in all cases) satisfactory (Table 2). Our representative cases are presented below to better understand the modified KPIF reconstruction in the scalp and forehead defects.

### 3.1. Case Presentations

#### 3.1.1. Case 1: Frontal Scalp Defect

A 76-year-old male patient presented with frontal scalp necrosis after a skin avulsion injury. Before flap surgery, wound preparation, including conventional wound dressing, debridement, and empirical antibiotic treatment, was performed for 1 week. Subsequently, we conducted a final debridement followed by KPIF reconstruction. The size of the final defect was 3.5 cm × 4 cm, and we covered the defect with a SMUM and OVC KPIF (6 cm × 12 cm) from the temporoparietal scalp and lateral forehead. One side of the donor site was primarily closed, and the other (lateral forehead) was closed with a skin graft (1.5 cm × 2 cm). The flap fully survived, and the skin graft at the donor site healed completely without postoperative complications. After a 10-month follow-up, the patient’s satisfaction score was 9, and the Harris 4-stage scale was rated 2 = excellent and 1 = good. Figure 2 shows the clinical photographs of case 1.

#### 3.1.2. Case 2: Vertex Scalp Defect

A 69-year-old male patient presented with vertex scalp necrosis after a skin avulsion injury. Before flap surgery, wound preparation, including conventional wound dressing, debridement, and empirical antibiotic treatment, was performed for 1 week. Subsequently, we conducted final debridement followed by flap coverage surgery. After debridement, the intact midportion of the original avulsion skin flap was used as a rotational flap for anterior defect (2 cm × 3.5 cm) coverage. The size of the final defect was 3 cm × 4 cm, and the defect was covered with hemi-KPIF (6 cm × 7 cm) from the occipital scalp. Direct closure of the donor site was then achieved. Complete survival of all flaps was observed without postoperative complications. After a 6-month follow-up, the patient’s satisfaction score was 9, and the Harris 4-stage scale was rated 1 = excellent and 2 = good. Figure 3 shows the clinical photographs of case 2.

#### 3.1.3. Case 7: Defect from the Frontal Scalp to the Lateral Forehead

An 84-year-old male patient experienced frontal scalp necrosis after a skin avulsion injury. Before flap surgery, wound preparation, including conventional wound dressing, debridement, and empirical antibiotic treatment, was performed for 2 weeks. Subsequently, we conducted a final debridement followed by KPIF reconstruction. The size of the final defect was 3.5 cm × 4 cm, and we covered the defect with double-opposing hemi-KPIFs (4.5 cm × 4.5 cm and 4.5 cm × 4.5 cm) from the frontal scalp and lateral forehead. Direct closure of all donor sites was achieved. All flaps survived, but marginal maceration of one donor site developed. The patient healed well with conservative management for <2 weeks. However, no other postoperative complications were noted. After a 9-month follow-up, the patient’s satisfaction score was 8, and the Harris 4-stage scale was rated good. Figure 4 shows the clinical photographs of case 7.

#### 3.1.4. Case 9: Temporal Forehead Defect

After a punch biopsy, a 58-year-old male patient was diagnosed with basal cell carcinoma of the temporal forehead. The lesion was excised with a 4-mm safety margin, and the final defect size was 4 cm × 4.5 cm. We covered the defect with a SMUM and OVC KPIF (6.5 cm × 12 cm) from the temporoparietal scalp and lateral forehead, and the donor site was directly closed. All flaps survived without postoperative complications. After an 8-month follow-up, the patient’s satisfaction score was 9, and the Harris 4-stage scale was rated good. Figure 5 shows the clinical photographs of case 9.

## 4. Discussion

We presented a single surgeon’s experience with scalp and forehead reconstruction with four modified KPIF techniques in 12 consecutive cases. We attribute our favorable outcomes to the adequate application of these KPIF modifications and other ancillary procedures.

The ideal reconstruction of scalp and forehead defects depends on the overall anatomy, a comprehensive understanding of reconstructive techniques, and a detailed assessment of each patient factor [1,2]. Furthermore, the scalp and forehead tissue quality are tauter and less elastic than other head areas because of their convexity [1,2,3,4]. In addition, moderate tension is inevitably encountered when closing and covering convex surfaces [1,2]. Figure 6 illustrates the anatomic regions of the scalp and forehead. In the scalp, the inelastic galea layer is responsible for the tight and loose areas of the scalp [2]. The galea is fully formed in the vertex scalp and blended into the temporoparietal and musculature fascia in the frontal, temporoparietal, and occipital scalps [2]. The skin is tight and inelastic in the vertex scalp; however, it has improved mobility and can be more easily rearranged in the periphery of the frontal, temporoparietal, and occipital scalps [2]. In the forehead, the central forehead, which is an extension of the scalp, is convex, thick, and somewhat inelastic, and its skin is tightly adherent to the underlying frontalis muscle [1,4]. In contrast, the lateral forehead and temple areas are somewhat concave and more elastic, and its skin is loosely attached to the underlying temporalis fascia [4]. In local flap reconstruction of the scalp and forehead, these loose and elastic areas frequently act as a good reservoir of tissue for reconstruction [1,2,3]. Therefore, reconstructive surgeons should design and elevate flaps in the areas closest to the defect when the local flap technique is used for scalp and forehead reconstruction. Following this basic principle, the flap was designed to be in an area with sufficient tissue laxity and elasticity beside the defect. For example, the frontal scalp defect was covered with the KPIF from the temporoparietal scalp and lateral forehead in case 1. The vertex scalp defect was covered with the KPIF from the occipital scalp in case 2. In addition, the defect between the central and lateral forehead was covered with the KPIF from the lateral and temporal forehead in case 6. The frontal scalp defect was covered with the KPIF from the frontal scalp and lateral forehead in case 7.

Several reconstructive algorithms have been developed for scalp and forehead defect coverage. Some factors should be considered, such as defect size, location, wound environment, tissue quality, and exposed structures, to perform successful local flap reconstruction in the scalp and forehead following these algorithms [1,2,3,4,20]. Local flap coverage is a good and useful option for small- to moderate-sized defects in the scalp and forehead [1,2,3,4,8,20]. The moderate-sized defect is defined as ≥30–40 cm and ≥20 cm in the scalp and forehead, respectively; however, there are no clear standards [2,3,20]. In this study, we successfully covered the scalp and forehead defects to approximately 21 cm and 18 cm, respectively, with the modified KPIF technique in combination with ancillary procedures. The flap should be roughly designed to be 2–6 times larger than the original defect to accommodate the lack of tissue elasticity in the scalp and forehead [2,3,4]. Therefore, we designed the flap width to be 1.5–2 times larger than the defect width. However, on the scalp and forehead, it is sometimes difficult to completely cover the defect, even if a single flap larger than the size of the defect is used. Hence, previous studies have recommended that surgeons should not hesitate to use multiple flaps in local flap reconstruction for scalp and forehead defect coverages [2,3,8]. We attempted to cover the defect with the modified KPIF as much as possible and performed ancillary procedures, such as skin grafting and other local flaps, to cover either the donor site or the remaining defect in five cases. It is important to perform the modified KPIF to cover the crucial area of the bone-exposed or deeper defect and to perform the above-mentioned ancillary procedures to cover the remaining superficial defect area. Regarding the surrounding tissue quality and wound environment, flap reconstruction should be performed after achieving proper wound preparation in non-oncologic defects and securing safety margins in oncologic defects. The flap should be harvested in the area away from the injury zone with sufficient tissue laxity [7,11]. This case-specific systemic approach guarantees ideal scalp and forehead reconstruction with a local flap technique.

The scalp and forehead have five anatomic layers as follows: the skin, subcutaneous connective tissue, aponeurosis (galea aponeurotica and temporoparietal fascia), loose areolar connective tissue, and pericranium (periosteum) [1,2,3,4]. Most local flaps in the scalp and forehead are elevated (harvested) via the subgalea plane, and movements to cover defects are generally achieved through transposition, rotation, and advancement [1,2,3,4]. The KPIF’s original design involves a curved-trapezoidal design consisting of two V-Y advancement flaps that move in the end-to-side direction and provide KPIF mobility [9]. The movement of the two V-Y advancement flaps results from the stepwise division of tissue layers, including the skin, subcutaneous tissue layer, and deep fascia layer, and minimal undermining of the flap margin [10,12]. In the scalp and forehead, the KPIF movement is generated by skin division, subcutaneous connective tissue, and aponeurosis. The less elastic tissue quality in the scalp and forehead requires more flap movement than in other areas, and the convexity of their underlying structures leads to substantial tension in wound closure. A modification that provides further flap movement and minimal wound tension are necessary for KPIF reconstruction of the scalp and forehead. Previous studies have verified the in vivo tension-reducing effect of KPIF reconstruction, mainly obtained by dividing the deep fascia layer [10]. We applied four modified KPIF techniques that entailed the division of the deep fascia (galea aponeurotica and temporoparietal fascia) in this study [11]. The hemi-KPIF devised by Petukhova et al. is a frugally modified type of KPIF in terms of less incision area and less morbidity [11,14]. It can provide a more increased flap mobility than the original KPIF through additional rotation movement and further undermining the flap margin [11,14]. We used this hemi-KPIF in five cases (including three cases of combined ancillary procedures and one case of double-opposing hemi-KPIF). The hemi-KPIF is identical to a Limberg flap but has some differences as follows: the former is designed at a long-axis corner of the defect parallel to the long axis, while the latter is designed at a short-axis corner of the defect vertical to the long axis; the former moves mainly advanced and additionally rotated, but the latter moves transposed and rotated. The SMUM KPIF devised by Moncrieff et al. provides further flap mobility through more undermining of the flap with safety because of structural stabilization via a maintained skin bridge along the flap’s outer curvilinear line [11,15,17]. The OVC KPIF also increases the flap mobility through additional rotation movement via closure with a fish-mouth fashion [11,16,17]. We used the combination of SMUM and OVC KPIF in four cases. This combination may be a convenient and safe KPIF modification for stabilizing flap mobility [11,17]. The modified Type II KPIF significantly increases flap mobility through the complete division of all surrounding tissue layers by forming the true island-form flap [10,11]. However, it should be used with caution in the tight scalp area (vertex scalp) because a measure of flap undermining, which is apt to impede flap perfusion, is frequently inevitable against inelastic galea aponeurotica. Therefore, we performed this modified Type II KPIF combined with the ancillary skin graft in one case of vertex scalp defect and used this modification alone in each case of the forehead and occipital scalp defects.

Despite our successfully modified KPIF reconstruction of the scalp and forehead, this study had some limitations. First, this study had a relatively low level of evidence because it is a retrospective review of cases with small sample size. Furthermore, the cases were heterogeneous, without a comparison group. In addition, our study has a relatively short follow-up period for evaluating postoperative scars because full scar maturation generally requires ≥12 months [11]. Therefore, future studies with prospective designs, larger sample sizes, appropriate comparison groups, and longer follow-ups are needed to validate the consistent outcomes for scalp and forehead reconstruction using modified KPIFs. Nonetheless, to the best of our knowledge, this study is the first single surgeon’s consecutive case series of scalp and forehead reconstruction using modified KPIF techniques. Furthermore, our study may assist reconstructive surgeons in planning the scalp and forehead reconstruction because we described each KPIF modification with representative case presentations in detail. Meanwhile, most KPIF reconstructions are possible under local anesthesia, which is advantageous in older and/or compromised patients. However, we performed the KPIF reconstruction under general anesthesia in this study because all patients did not accept local anesthesia and had no contraindications with the general anesthesia.

## 5. Conclusions

We successfully modified the KPIF reconstruction for small- to moderate-sized scalp and forehead defects. Our experience provides an extended application of KPIF reconstruction in plastic surgery. Based on our results, the four KPIF modifications (hemi-, SMUM, OVC, and modified Type II KPIF), either alone or with other reconstructive modalities, are good and reliable reconstructive options for covering defects in the scalp and forehead.

## Figures and Tables

**Figure 1 jpm-13-00329-f001:**
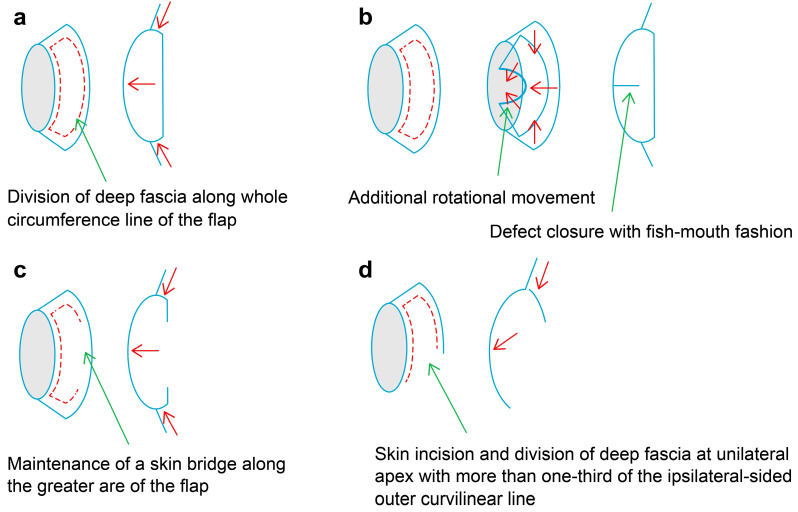
A basic illustration of the four modifications of the keystone perforator island flap (KPIF) used in this study. (**a**) Modified Type II KPIF. (**b**) Omega variation closure (OVC) KPIF. (**c**) Sydney melanoma unit modification (SMUM) KPIF. (**d**) Hemi-KPIF. Red dotted lines represent the division of deep fascia, and red arrows represent the direction of flap movement. (Reprinted from Keun Hyung Kim et al. [11], with permission from Hindawi).

**Figure 2 jpm-13-00329-f002:**
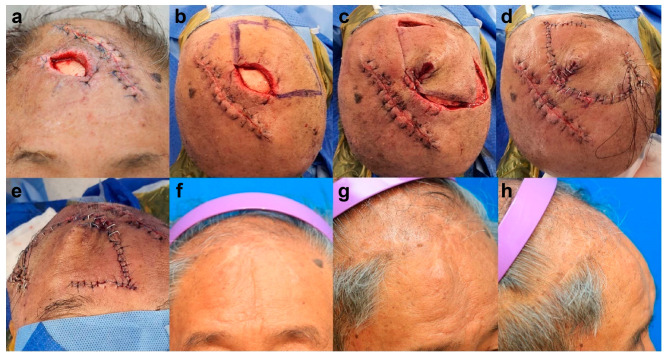
Clinical photographs of case 1. (**a**) Full-thickness skin defect (3.5 cm × 4 cm) with bone exposure in the frontal scalp area. (**b**) Design of a keystone perforator island flap (KPIF) (6 cm × 12 cm) in the temporoparietal scalp and lateral forehead. (**c**–**e**) Successful coverage of the defect with the Sydney melanoma unit modification and omega variation closure KPIF and skin graft (1.5 cm × 2 cm) for the lateral forehead donor site. (**f**–**h**) Postoperative photographs after a 10-month follow-up.

**Figure 3 jpm-13-00329-f003:**
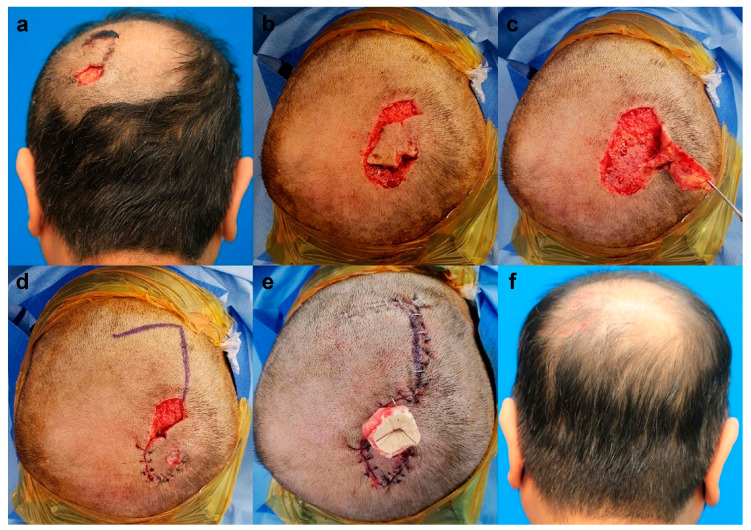
Clinical photographs of case 2. (**a**) Skin necrosis and defect in the vertex scalp after skin avulsion injury. (**b**,**c**) After debridement, the intact midportion of the original avulsion skin flap was used as a rotational flap for the anterior defect (2 cm × 3.5 cm) coverage. (**d**) Design of a hemi-keystone perforator island flap (KPIF) (6 cm × 7 cm) for the posterior final defect (3 cm × 4 cm) coverage in the occipital scalp. (**e**) Successful coverage of all defects with the rotational flap and the hemi-KPIF. (**f**) Postoperative photographs after a 6-month follow-up.

**Figure 4 jpm-13-00329-f004:**
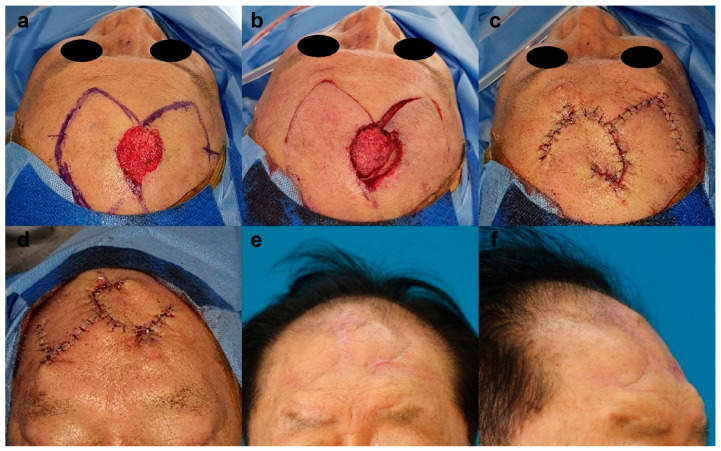
Clinical photographs of case 7. (**a**) Skin defect (3.5 cm × 4 cm) in the frontal scalp, and design of double-opposing hemi-keystone perforator island flaps (KPIFs) (4.5 cm × 4.5 cm and 4.5 cm × 4.5 cm) from the frontal scalp and lateral forehead. (**b**–**d**) Successful coverage of the defect with the double-opposing hemi-KPIFs. (**e**,**f**) Postoperative photographs after a 9-month follow-up.

**Figure 5 jpm-13-00329-f005:**
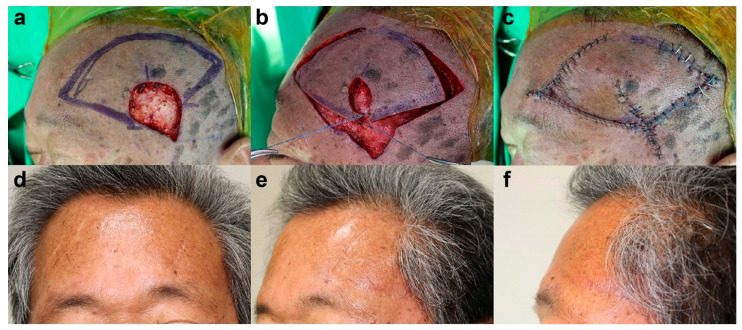
Clinical photographs of case 9. (**a**) Skin defect (4 cm × 4.5 cm) in the temporal forehead, and design of the keystone perforator island flaps (KPIF) (6.5 cm × 12 cm) from the temporoparietal scalp and lateral forehead. (**b**,**c**) Successful defect coverage with the Sydney melanoma unit modification and omega variation closure KPIF. (**d**–**f**) Postoperative photographs after an 8-month follow-up.

**Figure 6 jpm-13-00329-f006:**
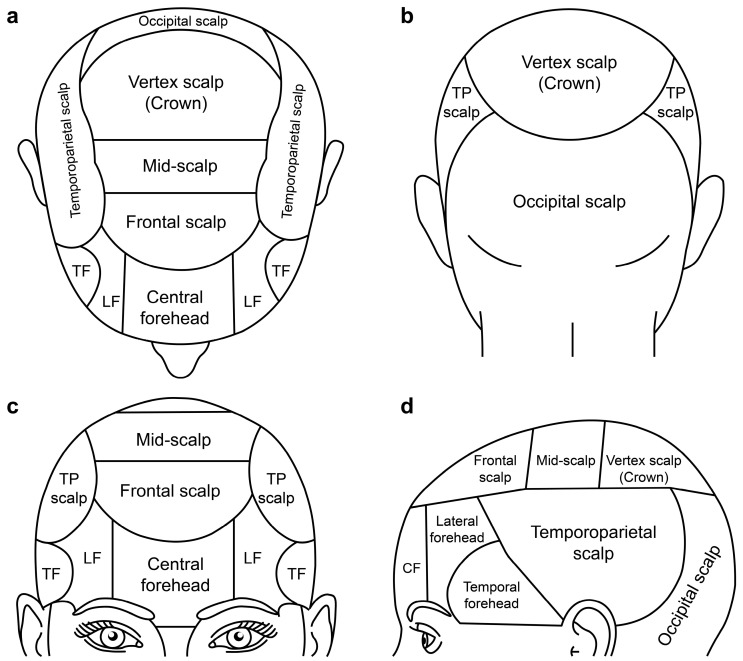
Schematic illustrations of anatomic regions in the scalp and forehead. (**a**) Top view of the head. (**b**) Posterior view of the head. (**c**) Anterior view of the head. (**d**) Lateral view of the head. TF, temporal forehead; LF, lateral forehead; TP, temporoparietal; CF, central forehead.

**Table 1 jpm-13-00329-t001:** Summary of patients’ data and characteristics.

Case no.	Sex/Age (yrs)	Defect Cause	Defect Location	Defect Size (cm × cm)	Flap Location	Flap Size (cm× cm)	Flap Type	Ancillary Procedures	Postoperative Complications	Flap Survival	Follow-Up Periods (Months)
1	M/76	Skin necrosis after skin avulsion injury	Frontal scalp	3.5 × 4	Temporoparietal scalp and lateral forehead	6 × 12	SMUM and OVC KPIF	Skin graft for donor site closure (1.5 cm × 2 cm)	None	Fully survived	10
2	M/69	Skin necrosis after skin avulsion injury	Vertex scalp	3 × 4	Occipital scalp	6 × 7	Hemi-KPIF	Rotational flap for remaining defect coverage (2 cm × 3.5 cm)	None	Fully survived	6
3	F/78	Skin necrosis after craniotomy	Occipital scalp	3.5 × 4	Occipital scalp	6 × 7	Hemi-KPIF	None	None	Fully survived	4
4	F/69	Skin necrosis after craniotomy	Frontal scalp	3 × 7	Temporoparietal scalp	6.5 × 7	Hemi-KPIF	Skin graft for donor site closure (2 cm × 2.5 cm)	None	Fully survived	5
5	M/77	Cellulitis resulted from a ruptured epidermoid cyst	Occipital scalp	3 × 4	Occipital scalp	4.5 × 10	SMUM and OVC KPIF	None	None	Fully survived	6
6	M/76	Skin necrosis after skin avulsion injury	Central and lateral forehead	2.5 × 5	Lateral and temporal forehead	3.5 × 4	Hemi-KPIF	Skin graft for remaining defect coverage (1.5 cm × 3 cm)	None	Fully survived	10
7	M/84	Skin necrosis after skin avulsion injury	Frontal scalp	3.5 × 4	Frontal scalp and lateral forehead	4.5 × 4.5, 4.5 × 4.5	Double-opposing hemi-KPIF	None	Marginal maceration at the 3-point area of donor site → Conservative management	Fully survived	9
8	M/78	Wide excision of basal cell carcinoma	Frontal scalp	3 × 4.5	Temporoparietal scalp and lateral forehead	4.5 × 9.5	SMUM and OVC KPIF	None	None	Fully survived	9
9	M/58	Wide excision of basal cell carcinoma	Temporal forehead	4 × 4.5	Temporoparietal scalp and lateral forehead	6.5 × 12	SMUM and OVC KPIF	None	None	Fully survived	8
10	F/63	Wide excision of basal cell carcinoma	Lateral forehead	2 × 2	Lateral forehead	3.5 × 7	Modified Type II KPIF	None	None	Fully survived	10
11	M/68	Skin necrosis after skin avulsion injury	Vertex scalp	3 × 5	Temporoparietal scalp	7 × 16	Modified Type II KPIF	Skin graft for remaining defect coverage (1 cm × 3.5 cm)	None	Fully survived	6
12	M/67	Cellulitis resulted from a ruptured epidermoid cyst	Occipital scalp	2.5 × 3.5	Occipital scalp	4 × 11	Modified Type II KPIF	None	None	Fully survived	9

no., number; yrs, years; M, male; F, female; SMUM, Sydney melanoma unit modification; OVC, omega-variation closure; KPIF, keystone perforator island flap.

**Table 2 jpm-13-00329-t002:** Outcomes of the final scar appearance were evaluated using the patient satisfaction surveys and Harris 4-stage scale by three independent plastic surgeons.

Case no.	Patient Satisfaction Score (Scale, 1–10)	PS1	PS2	PS3
1	9	Excellent	Excellent	Good
2	9	Excellent	Good	Good
3	8	Excellent	Good	Good
4	8	Good	Good	Good
5	8	Good	Excellent	Good
6	7	Good	Good	Good
7	8	Good	Good	Good
8	8	Good	Good	Excellent
9	9	Good	Good	Good
10	8	Good	Good	Good
11	7	Fair	Good	Fair
12	9	Excellent	Good	Excellent

PS, plastic surgeon; no., number.

## Data Availability

The data presented in this study are available on request from the corresponding author. The data are not publicly available due to privacy restrictions.

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
