# Peer review of "Modified Keystone Perforator Island Flap Techniques for Small- to Moderate-Sized Scalp and Forehead Defect Coverage: A Retrospective Observational Study"

_jpm, 2023, doi:10.3390/jpm13020329_

Round 1

Reviewer 1 Report

The manuscript offers the authors’ personalized view on KPIF use for scalp and forehead reconstruction by performing some flap modifications according to experience. The reconstructive technique is well illustrated by drawings and clinical cases and could benefit the practice of reconstructive surgeons. Although a small case series is included, the provided data has useful clinical information. The study design is well suited to reach the desired purpose and some objective means of assessing the postoperative outcomes were used.

Minor language corrections or rephrasing are necessary for better understanding of the intended meaning of the text: in line 39 ‘to which’ should be replaced with ‘in which’. Similarly, in lines 43, 44 the correct meaning of the text would be ‘to cover areas with exposed alloplastic materials and neurocranial structures’.

Author Response

Response to Reviewer 1 comments

<Reviewer 1>

The manuscript offers the authors’ personalized view on KPIF use for scalp and forehead reconstruction by performing some flap modifications according to experience. The reconstructive technique is well illustrated by drawings and clinical cases and could benefit the practice of reconstructive surgeons. Although a small case series is included, the provided data has useful clinical information. The study design is well suited to reach the desired purpose and some objective means of assessing the postoperative outcomes were used.

Minor language corrections or rephrasing are necessary for better understanding of the intended meaning of the text: in line 39 ‘to which’ should be replaced with ‘in which’. Similarly, in lines 43, 44 the correct meaning of the text would be ‘to cover areas with exposed alloplastic materials and neurocranial structures’.

Response: We would like to thank Reviewer 1 for the time and effort in reviewing our manuscript and providing comments and suggestions, which have considerably helped us improve our manuscript. We have responded to your comments below and hope our responses and revisions address all your comments.

As you mentioned, we have revised the relevant contents in lines 39 and 42–44, page 1 as follows:

“Flap surgery is essential for adequate coverage of deep defects where the underlying structures are exposed [6].”

“Therefore, the free flap technique is the best reconstruction approach, particularly when there is a need to cover areas with exposed alloplastic materials and neurocranial structures [8].”

Additionally, the manuscript has undergone another English editing process, and we have provided the certification of the English editing service by the Editage as a separate file.

Reviewer 2 Report

The authors present  a series of KPIF (keystone perforator island flap dor small to moderate-sized skalp defects. 

These are intersting results as an alternative but with only few patients. The authors should correct  "Defect size" in text and tables - not in cm², these are dimensions in cm. 

I our opinion, most of theses KPIF are possible under local anaesthesia, one great adavnmtage in older /and/or compromised patients!

Author Response

Response to Reviewer 2 comments

<Reviewer 2>

The authors present a series of KPIF (keystone perforator island flap for small to moderate-sized scalp defects.

These are interesting results as an alternative but with only few patients. The authors should correct "Defect size" in text and tables - not in cm², these are dimensions in cm.

Response: We would like to thank Reviewer 2 for the time and effort in reviewing our manuscript and providing comments and suggestions, which have considerably helped us improve our manuscript. We have responded to each of your valuable comments below and hope our responses and revisions address all your comments.

Accordingly, we have revised all dimensions of the defect and flap size to cm in the manuscript.

In our opinion, most of these KPIF are possible under local anesthesia, one great advantage in older /and/or compromised patients!

Response: Thank you for this valuable comment. According to your advice, we have described the related contents in the Discussion section as follows (page 12; lines 345–349):

“Meanwhile, most KPIF reconstructions are possible under local anesthesia, which is advantageous in older and/or compromised patients. However, we performed the KPIF reconstruction under general anesthesia in this study because all patients did not accept local anesthesia and had no contraindications with the general anesthesia.”

Reviewer 3 Report

Dear Editor.

I enjoyed reading the article entitled: "Modified Keystone Perforator Island Flap Techniques for 2 Small- to Moderate-Sized Scalp and Forehead Defect Coverage: 3 A Retrospective Observational Study".

Overall, the article is well written and the results well presented.

Even if I am not familiar with the techniques described, I found figure 1 not very clear.

Can it be made or described more clearly?

Otherwise I find it well written and well presented.

I certainly think it is possible to improve English in general.

Author Response

Response to Reviewer 3 comments

<Reviewer 3>

I enjoyed reading the article entitled: "Modified Keystone Perforator Island Flap Techniques for 2 Small- to Moderate-Sized Scalp and Forehead Defect Coverage: 3 A Retrospective Observational Study".

Overall, the article is well written and the results well presented.

Even if I am not familiar with the techniques described, I found figure 1 not very clear.

Can it be made or described more clearly?

Otherwise I find it well written and well presented.

I certainly think it is possible to improve English in general.

Response: We would like to thank Reviewer 3 for the time and effort in reviewing our manuscript.

We have included a larger version of Figure 1.

Additionally, the manuscript has undergone another English editing process, and we have provided the certification of the English editing service by the Editage as a separate file.